# Causes of Stress among Healthcare Professionals and Successful Hospital Management Approaches to Mitigate It during the COVID-19 Pandemic: A Cross-Sectional Study

**DOI:** 10.3390/ijerph191912963

**Published:** 2022-10-10

**Authors:** Lourdes Herraiz-Recuenco, Laura Alonso-Martínez, Susanne Hannich-Schneider, Jesús Puente-Alcaraz

**Affiliations:** 1Cardiovascular Research Institute of Basel, Spitalstrasse 2, CH-4056 Basel, Switzerland; 2Department of Health Science, University of Burgos, Paseo de los Comendadores, s/n, 09001 Burgos, Spain; 3Klinikum Mittelbaden, Dr.-Rumpf-Weg 7, 76530 Baden-Baden, Germany

**Keywords:** COVID-19, hospital occupational health, management, mental stress, healthcare professionals

## Abstract

The outbreak of the COVID-19 pandemic posed an immediate challenge to the management of hospitals in Germany and elsewhere. The risk of stress for front-line healthcare professionals forced occupational health and safety units to adopt a variety of protective measures, not all of which have been thoroughly validated. The main objective of the present analysis is to assess what the most important sources of stress were and which of the protective measures applied to counteract stress among healthcare staff had the greatest impact. A better understanding of these factors will improve hospital management and worker safety in a future health crisis situation and may also prove to be beneficial in non-crisis situations. For this purpose, in 2020, an exploratory, cross-sectional and quantitative study using a questionnaire created for this purpose was carried out on a total of 198 professionals—133 nurses and 65 physicians—at the Klinikum Mittelbaden Balg hospital in Baden-Baden, Germany, during the first wave of the pandemic. Statistical analyses showed that nurses suffer more stress than physicians and that stress is higher among professionals in critical care and emergency units than in units that are less exposed to infected patients. It was also found that measures such as salary incentives, encouragement of work in well-integrated teams, and perceived support from hospital management mitigate stress. These findings highlight the importance of support measures from management and superiors. Knowing the actual effectiveness of the measures applied by management and the factors mentioned above could help to protect healthcare professionals in the event of another pandemic or similar situations and may still be of value in dealing with the continuing COVID-19 pandemic.

## 1. Introduction

On 11 March 2020, the World Health Organisation (WHO) confirmed the existence of the COVID-19 pandemic, caused by SARS-CoV-2 [1]. The high transmission rate of the virus and its rapid spread between countries forced governments and key authorities to implement strict health and isolation measures [2].

The unexpected, sudden and unknown nature of the COVID-19 pandemic has had an impact at all levels, generating unprecedented health, economic and social crises. Both the social distancing measures (home confinement and quarantine) and the severe worldwide slowdown of economic activity are situations whose long-term consequences are unknown and unpredictable [3].

Beyond the risk of viral infection, COVID-19 also posed a risk to the mental health of frontline professionals. While the rest of the population reduced their exposure to infected individuals, these professionals worked in direct interaction with infected patients and were subject to a greater risk of infection themselves [4].

Shortages of materials and safety equipment, a high demand for hospital services, and the reduced availability of trained personnel posed a complex environment and increased the fear of infection [5,6]. Some studies revealed a high psychological impact of this threatening environment, indicating that the stress level of healthcare workers increased during the pandemic, along with other psychological problems, such as depression, anxiety and insomnia [6,7,8,9].

In December 2020, the WHO published “Health workforce policy and management in the context of the COVID-19 pandemic response: interim guidance”, which addresses issues such as adequate working conditions, including occupational health and safety, as well as the mental well-being of the health workforce [10]. The WHO guidance prioritises the mental well-being of the health workforce in order to ensure both the long-term staffing levels and the short-term crisis response. Its first recommendation for hospital management is to assess and minimise the added psycho-social risks of occupational stress related to COVID-19, including the increased likelihood of being subject to discrimination, violence, harassment, and stigmatisation as a result of the increased risk of infection inherent in working with infected patients [10,11,12].

In Germany, the health management services followed the recommendations of national and international organisations, making good use of available resources. Beginning in 2021, the federal and regional governments approved a special payment called the “Corona bonus” as a reward for having borne an increased workload and added risk during the pandemic. In addition, some hospitals, including the centre where this study was conducted, used their own funds to provide extra salary incentives to their staff, also called “Corona bonuses.”

To understand how a health crisis influences the mental health of the population, stressors can be classified into three levels: macro, meso, and micro. In this classification scheme, the macro level corresponds to socio-political and cultural factors, the meso level to organizational factors and the availability of resources, and finally, the micro level refers to interpersonal relationships and socio-psychological factors [13].

In the context of the current pandemic, it was necessary to measure its impact on the mental health of the healthcare workforce almost in real time, in order to determine the degree of stress and to assess whether improvised preventive measures were effective. The vast majority of healthcare professionals adapted to face the challenges caused by the coronavirus pandemic, but they continued to face heightened levels of stress in their work [7].

The main problems that accompanied their daily work were the high the demand for care due to the lack of personnel available during sick leave or vacations, which also had to be deferred or cancelled during the pandemic. Postponing vacations added another stressor and took away a major means to recover from extended periods of stress. Other problems were the fear of contracting the disease and transmitting it to people close to them, the availability of equipment that is not very comfortable and limits mobility, and the lack of psychological assistance for patients and relatives [7].

Additional factors were the exposure to grief and the difficulty of decision making in situations with an uncertain prognosis, which many professionals were confronted with in a short time [14]. As a result, healthcare professionals continued to face situations of emotional (anxiety, frustration, or guilt), behavioural (uncontrolled crying, isolation, or difficulty with self-care), cognitive (confusion, intrusive thoughts or difficulty making decisions), and physical (tremors, headaches, insomnia, exhaustion or altered appetite) stress [15,16]. In order to improve the mental health of healthcare workers, it was essential to implement plans that focused on the constant evaluation of their mental health to allow for early detection and intervention. 

Taking into account all the above problems and the need to act quickly and effectively, ad hoc instruments created to measure variables in specific situations became essential in an effort to determine which factors and problems affected mental health and in order to develop protocols to reduce stress and to plan coping strategies.

Measurement of the psychological impact of the COVID-19 pandemic can follow one of two basic strategies. One possibility is to use pre-existing, generic tools for the assessment of stress and mental health, the other is to develop new tools adapted to the peculiarities of this pandemic. 

Examples of the first approach include the PHQ-9 (Patient Health Questionnaire) for the diagnosis of depression, the Generalised Anxiety Disorder Scale (GAD-7), the Perceived Stress Scale (PSS-10-C), or the Impact Event Scale (IES), which assess subjective anxiety derived from any life event [16,17,18]. In some cases, items were combined from existing scales and additional socio-demographic variables were incorporated to obtain more specific results [18]. This was the case in a study conducted in Iran on 60 healthcare professionals to validate a new tool, which combined a depression, anxiety, and stress scale (DASS-21) created by Lovibond in 1995, and six of the items that compose the Copenhagen Burnout Inventory (CBI) [19,20]. Others applied instruments used in similar circumstances to the current pandemic, such as the SARS-based Caring for Patients with a Highly Infectious Disease Stress Scale developed by Baoyu Zhuang in 2005 [21,22].

We chose the second ad hoc approach, even though prior validation was impossible, in order to better characterize this exceptional situation. In order to prevent mental stress and implement effective measures to improve the situation of healthcare workers, an ad hoc approach allows for a more focused measurement of the parameters of interest and the use of more contextualised questions relevant to the COVID-19 pandemic [15]

The management team of the Klinikum Mittelbaden Balg general hospital in Baden-Baden, in the federal state of Baden-Würrttemberg, Germany, realised there was a need to design a new tool to assess which factors generate stress and which of the implemented measures reduced it.

In the search for solutions, not altering holidays or improving shift rotation are often proposed as effective measures to mitigate stress. However, these measures are often unrealistic in a pandemic situation, where there is a chronic shortage of healthcare staff, leaving management no other choice but to defer holidays in order to ensure sufficient staffing levels [23].

In the rapidly evolving pandemic, the initial intent of the questionnaire was to determine stress factors triggered by the pandemic conditions and to propose additional actions necessary to improve occupational safety conditions that may not initially have been taken into account or which had not been properly implemented. After the surveys were carried out, an initial descriptive assessment was made and the data were archived. However, a subsequent review of the data suggested that the full potential of the analysis had not been achieved, which provided the basis for this study. 

The questionnaire not only includes factors dependent on hospital management (meso level), such as the management of human and material resources, but also analyses factors not dependent on hospital management, such as the perceived socio-political appreciation, the burden of lockdown (macro level) or the disruption of interpersonal relations in the social and family environment (micro level). These factors lead to absenteeism, sickness, or burnout among healthcare workers, so knowing their impact helps to improve management with regard to family conciliation strategies or shift work scheduling, which needs to be improved; however, their improvement during a pandemic situation poses an even greater challenge.

We found the questionnaire was consistent and reliable for estimating stress during the pandemic outbreak, during which the mental health risk was very high. As a result, we were able to determine which factors are aggravating and which are protective against stress among nursing and medical staff working in the different care services for patients with COVID-19. The addressed hypotheses are: 

**H1:** 
*Fear of spreading the disease in the immediate environment, difficulties in accessing personal protective equipment, changes introduced during the pandemic in work procedures and organisation*
*, and perceived social stigma of working in direct contact with people with COVID-19 contribute to higher levels of stress among medical staff.*


**H2:** 
*Incentive payments, specific support from health management, social recognition for the work done*
*, and teamwork are protective factors against stress in the healthcare workforce.*


**H3:** 
*There are differences in the impact of and response to the implemented management measures between nurses and physicians, as well as among the different care units for patients with COVID-19.*


## 2. Materials and Methods

### 2.1. Design

An exploratory, descriptive, observational, quantitative, and cross-sectional study was conducted to assess the relationship between different stressors and protectors of stress and the mental burden on the healthcare workforce in a German hospital during the first months of the COVID-19 pandemic. To this end, a questionnaire with specific items adapted to the healthcare context of the current pandemic was designed and carried out.

### 2.2. Participants

Of all the hospital’s healthcare workers from the Klinikum Mittelbaden Balg hospital in Baden-Baden (Baden-Württemberg, Germany), a total of 291 professionals, who had been subjected to high levels of mental stress during the first peak of the COVID-19 epidemiological outbreak were selected via non-random convenience sampling. All survey participants worked as part of the staff in one of the COVID-19 patient care services between March and June 2020.

### 2.3. Procedures and Instruments

The occupational medicine department of the Klinikum Mittelbaden Balg hospital developed an ad hoc questionnaire to assess the psychological burden during the first wave (March–June 2020) of the COVID-19 pandemic. In addition, the hospital’s IT department contributed to the development of the structure of the survey in order to facilitate its application and interpretation.

The survey items were formulated taking into account the new paradigm that had emerged around the COVID-19 pandemic. Hence, it was considered important to divide the questionnaire into two sections: the first to find out what problems were affecting workers in different areas and the second to evaluate the newly implemented measures intended to reduce stress and other unknown risks.

During July and August of 2020, printed copies of the questionnaire and an informed consent form were distributed to the COVID-19 patient care units: the outpatient department (tent) for COVID-19 patient screening at emergency doors, the emergency department (ED), the intensive care unit (ICU), and the COVID-19 hospital ward. The survey was distributed in German and has been translated into English for this manuscript. 

Finally, approval was requested from the Bioethics Committee of the University of Burgos, which was granted (REF. No. UBU 016/2021). The occupational medicine department provided the questionnaires in paper format, which were thoroughly checked [24].

As mentioned above, the questionnaire was divided into two scales: the first one, called the Stress Scale, initially comprised 13 items (Table 1) that evaluate different stressors; the second scale, called the Satisfaction Scale, consisted of 10 items (Table 2) with different factors that are considered protective against the stress produced by the COVID-19 pandemic. Each item in the questionnaire was numbered using the acronyms of each scale—MS**T**DCP (Mental Stress during COVID-19 Pandemic) for the Stress Scale and MS**A**DCP (Mental Satisfaction during COVID-19 Pandemic) for the Satisfaction Scale.

As a result of the geographical location of the hospital, some of the staff reside in France. The border was closed for considerable periods of time prior to the survey, border checks were still in place, and the threat of renewed closure was always present. Therefore, the MS**T**DCP12 item was included in the first Stress Scale, which assessed commuting across the border as a stressor. Item MS**T**DCP13 analysed how stress resulted from the need to go to work, but having to stay at home in quarantine due to contact with a COVID-19-positive person. Since both questions were only asked of those who had these experiences and the number of responses were very low for both, they were excluded from the statistical analysis. Thus, the Stress Scale was composed of a total of 11 questions.

The responses were structured on a Likert-type scale ranging from 1 to 8. While a 5-point Likert-type scale would have been simpler, an 8-point scale widens the range of the responses to further lateralise them, avoids neutrality and reduces bias. In addition, the respondents had the option of writing an explanation their responses to each item. Subsequently, 5 score ranges were established to facilitate the interpretation of the results (Figure 1 and Figure 2).

In the case of the Stress Scale, 1 indicates not at all stressed, while 8 indicates extremely stressed. A score of 11 indicates very little or no stress, 12 to 37 low-stress levels, 38 to 62 moderate stress levels, 63 to 87 high stress levels, and a total score of 88 is considered extreme stress. 

For the Satisfaction Scale, a value of 1 indicates that respondents are very satisfied and a value of 8 indicates that respondents are not at all satisfied. A score of 10 indicates extreme satisfaction, from 11 to 34 high satisfaction, 35 to 57 moderate satisfaction, 58 to 79 low satisfaction, and a final score of 80 points indicates that respondents are not at all satisfied.

As mentioned above, in addition to assessing the stress of the healthcare workforce and the actions or factors that can mitigate stress, we evaluated whether the tool developed for this purpose is reliable. To this end, Cronbach’s α was calculated and found to be consistently above 0.8, where values > 0.71 show good internal consistency [25].

### 2.4. Statistical Analysis

The data from the surveys were input and analysed using the IBM SPSS v.26 statistical analysis software and the PROCESS v3.1 macro for IBM SPSS v24 [24] in order to conduct mediation and moderation analyses.

Before performing the different statistical analyses, the data were verified for normal distribution. Since the sample consisted of more than 50 subjects, the Kolmogorov–Smirnov (K-S) test was used, with D_(176)_ = 0.045, *p* > 0.05 for the Stress Scale and D_(124)_ = 0.075, *p* > 0.05 for the Satisfaction Scale. The obtained values show that the samples are normally distributed, so parametric tests were used to test the hypotheses with the selected variables [25].

In order to test the relationship between variables, Student’s *t*-test and ANOVA were employed to compare independent samples and Pearson’s correlation was used to assess differences between variables. ANCOVA was also used to eliminate the heterogeneity of certain items influenced by covariates. Multiple regression analysis was used to adjust linear models within the scores of relevant items with more than one independent variable. In addition, moderation and mediation analyses between different variables were carried out to observe the positive effects of protective stressor factors on the stressor factors. 

As no previous validation had been carried out for either scale, and they were applied to a sample with high levels of stress due to an exceptional situation, we carried out an Exploratory Factor Analysis (EFA) of these scales. This allows for the identification of factors that explain the configuration of correlations within a set of observed variables [26].

As for the *p*-values, values below 0.05 were taken as significant and those below 0.01 were considered highly significant [26] The effect of sample size was calculated based on Student’s *t*-test for independent samples, using Hedges’ g-value, which considers that values less than 0.2 indicate a small effect size, values less than 0.5 a medium effect size, and those less than 0.8 a large effect size [25].

The results of the ANOVA test were interpreted using the Eta squared value (ηp^2^), in which ηp^2^ = 0.01 indicates a small effect, ηp^2^ = 0.06 is a medium-size effect, and ηp^2^ = 0.14 is a large effect of the sample size [27].

The values of the correlations are considered small from ± 0.10 to ± 0.29, medium from ± 0.30 to ± 0.49, and large from ± 0.50 to ± 1.00 [27].

## 3. Results

The total study population was N = 291, of which 192 were nurses (87% female) and 99 were physicians (50.5% female). Of these, 133 nurses and 65 physicians (n = 198) responded to the questionnaire; the participation rate for nurses was 69.27% and 65.65% for physicians. Of the total number of respondents, 89.34% of the nurses were female, and 66.33% of the physicians were female.

### 3.1. Reliability and Validity

To initially explore the items that constitute the questionnaire, exploratory factor analysis (EFA) of both sections was carried out to confirm the factor structure grouping of items according to their characteristics. In addition, to verify whether the items had been correctly formulated, the qualitative data provided in the responses of the surveyed professionals were also considered. These comments served as indicators of quality and specificity and helped us to make relevant changes in order to optimise the assessment tool (Table 3 and Table 4).

According to the recommendations of del Líbano et al. and Wayne et al. [25,28], in the item selection process, items that differentiate extreme groups or that have item-total correlations higher than 0.30 should be kept, while the others should be removed. According to the data shown in Table 5 and Table 6, all the correlations and corrected scores were higher than 0.30 for all items, so all items were retained, as there were significant correlations with the total number of items.

Table 5 and Table 6 demonstrate that all item-total correlations exceed the recommended threshold of 0.30 and thus none of them were removed from the analysis [25,28].

#### 3.1.1. Exploratory Factor Analysis

Testing the initial set of items is important to build a suitable instrument for the measurement of this construct. For this purpose, a reliability analysis and external validation were carried out.

The internal consistency of the scale was obtained through the calculation of Cronbach’s alpha, which shows high reliability and internal consistency, with a value of α = 0.87 [28]. The mean score for the Stress Scale is 50.23 (*N* = 176, *SD* = 14.63) and the mean score for the Satisfaction Scale is 29.35 (*N* = 124, *SD* = 12.84). 

The Bartlett sphericity test on both scales was performed and the Kaiser, Meyer, and Olkin (KMO) sample adequacy measurement index was calculated, as well as the significance levels by principal components and Varimax rotation with Kaiser normalisation. These analyses are adequate to establish differences in the structure if there are underlying scales [15]. The minimum factor saturation limit was set at 0.40, below which loads would have been excluded, had we found such loads. Both the Stress Scale and the Satisfaction Scale had KMO values above the threshold of 0.75, namely 0.830 and 0.869, respectively, and Bartlett’s test was significant in both cases (*p* = 0.001), which provided the necessary prerequisites for a meaningful EFA because there were significant relationships between the items. The EFA was carried out to explore the set of latent variables or common factors that explain the responses to the items of these scales [23]. This allows for the determination of the dimensions and structure contained in the original variables.

Subsequently, bivariate correlations between items were considered to observe the existent relationships between them [28].

#### 3.1.2. Exploratory Factor Analysis (EFA) of the Stress Scale

The EFA of the items making up the Stress Scale resolved three component factors. The first component explained 38.40% of the variance of the Stress Scale, the second component 11.30%, and the third one 10.39%, together explaining 60.10% of the variance.

As can be seen in Table 7, the correlations obtained between each item and the corrected scores of the scale were higher than 0.30 in all cases and were thus retained in the analysis [25]. Taking into account the tests and statistical parameters described above, the resulting component matrix is adequate, though three items saturate above 0.30 in two factors (Table 7).

The factor structure obtained for the Stress Scale largely corresponds to three dimensions with good and acceptable reliabilities (Table 8 and Table 9).

#### 3.1.3. Exploratory Factor Analysis of the Satisfaction Scale

The EFA of the Satisfaction Scale items resolved two component factors. The first component explained 49.06% of the variance of the Satisfaction Scale and the second component explained 10.89%, together explaining 59.95% of the variance. The resulting component matrix complies with the established statistical parameters, although it is true that three items saturate above 0.30 in two factors (Table 10).

We found that by adding the MS**A**DCP8 item to the second component, the internal consistency reliability (α) of the second component increased, without relevant changes in the reliability of the first component (see Table 10 and Table 11).

The factor structure obtained for the Satisfaction Scale corresponds to two dimensions with good and acceptable reliabilities. The *α* value obtained for Factor 1 of the Satisfaction Scale is high and for Factor 2 it is acceptable (see Table 11 and Table 12).

### 3.2. Construct-Related Validity

The results shown in Table 13 demonstrate the existence of high correlations between the scales and their sub-dimensions, medium correlations between the two scales and low-to-medium correlations between the sub-dimensions of the different scales. These findings indicate that the higher the level of stress (higher score on the Stress Scale), the lower the satisfaction with management (higher score on the Satisfaction Scale).

### 3.3. Comparison of Means and Nominal Variables of the Scale

A comparison between groups according to the variables composing the questionnaire and its two subcomponent scales was made to evaluate the proposed hypotheses. The differences in the responses of the two professional groups were analysed using Student’s *t*-test.

There were statistically significant differences between the obtained means of the two groups, with the *p*-value being a statistically significant 0.0001. The means for nurses (*N* = 116) = 53.06 (*SD* = 14.59) and physicians (*N* = 60) = 44.77 (*SD* = 13.17) show that stress levels were higher in nursing staff. The sample size effect calculated using Hedges’ g-value for the Stress Scale (0.58) indicated a medium-sized effect.

Concerning the Satisfaction Scale, statistically significant differences *p* = 0.023 were also obtained, with a higher mean for nurses (*N* = 77) = 31.38 (*SD* = 13.56) than for physicians (*N* = 47) = 26 (*SD* = 11.49), meaning that job satisfaction and therefore the protective effect against stress was higher for nurses than for medical staff. The sample size effect calculated using Hedges’ g-value for the Satisfaction Scale was 0.42, indicating a medium-sized effect.

According to the data from the tests of inter-subject effects and observing the obtained means, the items in which differences between nurses and physicians are most significant are appreciation of professional experience (MSADCP3) (*p* = 0.001), the attention provided by occupational medical services (MSADCP5) (*p* = 0.006), satisfaction with crisis management (MSADCP1) (*p* = 0.009), and finally, perceived public support (MSADCP4) (*p* = 0.015).

An ANOVA (analysis of variance) test was performed to analyse the differences in the means according to the variables of the department for both scales. For the Stress Scale, a *p*-value = 0.053 indicates that there are potentially significant differences between the departmental stress means.

As shown in Table 14, the means varied from department to department, with the highest stress levels found among those working in the emergency department and the lowest among those working in the outpatient tent. The Bonferroni test confirmed the statistically significant difference (*p* = 0.053) between the emergency and outpatient tent, though there was no statistically significant difference between the emergency department and the ICU or COVID ward.

On the other hand, a *p*-value of 0.057 was obtained for the Satisfaction Scale, so there were significant differences between the means of the departments with regard to stress-protective factors. As shown in Table 15, the means varied from department to department, with the highest satisfaction levels found among those working in the outpatient tent and the lowest among those working in the emergency department. The Bonferroni test showed a trend-significant difference *p* = 0.069 between emergency department and the COVID ward/station, with the stress-protective effects being greatest for staff on the COVID ward.

Multiple regression analysis made it possible to analyse whether pandemic management (MS**A**DCP1) and the temporary lack of Personal Protective Equipment (PPE) (MS**T**DCP6) acted as predictors of the fear of COVID-19 infection (MS**T**DCP1). The ANOVA test resulted in a *p*-value of 0.016, so the regression analysis showed that there is an effect between the variables. Positive Beta values can be observed in the coefficient table, indicating that the higher the scores for MS**A**DCP1 and MS**T**DCP6, the higher the stress due to the fear of SARS-CoV-2 infection. The corrected R-squared value (*F*_(2,185)_ = 4.20, *p* = 0.0001, *R* = 0.209; *R*^2^ = 0.04; *Rcorrected*^2^ = 0.03) shows that only 3.3% of the variance of the fear of COVID-19 infection is explained by factors related to management and the lack of PPE.

Similarly, in order to observe the influence of the fear of infection (MS**T**DCP1), the fear of transmission (MS**T**DCP2), and avoidance by others (MS**T**DCP8) as predictors of greater stress due to extra private burdens (MS**T**DCP11), multiple regression analysis was performed, reaching a high level of significance (*p* = 0.0001). Thus, the regression analysis showed that there is an effect between the variables. In the table of coefficients, positive Beta values can be observed, indicating that the higher the scores (higher stress), the higher the private burden. According to the corrected *R-squared* value (*F*_(3,191)_ = 17.82, *p* = 0.0001, *R* = 0.47; *R*^2^ = 0.22; *Rcorrected*^2^ *=* 0.21 and the positive Beta values observed in the coefficient table, 20.6% of the variance of stress due to the increase in the private burden (MS**T**DCP11) is explained by the selected variables. 

### 3.4. Moderation and Mediation between Variables

The selection of variables for moderation and mediation analyses was based on our formal hypotheses mentioned above, prioritising the effect of pandemic management (MS**A**DCP1) (X) over the fear of infection (MS**T**DCP1) (Y) and whether the effect between these variables was moderated by the occupation (nurses/physicians) (W). The effect *b1* = 0.76, *SE* = 0.26, *p* = 0.0045 is significant, indicating that poor pandemic management increases the fear of SARS-CoV-2 infection. For *b2* = 1.21, SE = 0.64, *p* = 0.0598, the effect is significant, so there are differences in the fear of infection between nurses and physicians.

Finally, the coefficient *b3* = −0.41, *SE* = 0.19, *p* = 0.029 indicates a moderating effect of the variable occupation, so the effect of satisfaction with pandemic management over the fear of COVID-19 infection is dependent on the respondent’s profession. The effects of pandemic management on the fear of infection for nurses *b3*(1) = 0.35, SE 0.11, *p* = 0.0014 were positive and significant, whereas for physicians *b3*(2) = −0.06, SE = 0.15, *p* = 0.7118 they were not significant; thus, for nurses, greater satisfaction with pandemic management predicts a reduction in the fear of COVID-19 infection. Therefore, the hypothesis of the existence of differences in the impact and response to the implemented management measures between nurses and physicians, positing that the respondent’s profession modulates the relationship between their satisfaction with pandemic management and fear of COVID-19 infection, was confirmed (Figure 3). 

The mediation analysis showed whether satisfaction with extra financial compensation—in this case, an incentive payment or “Corona bonus” (MS**A**DCP2)—mediated the perception of perceived support from superiors (MS**A**DCP7) and the perceived occupational workload (MSTDCP9). The *a* coefficient was significant and positive (MSADCP7_support_ = 0.23, *SE* = 0.05, *p* = 0.0001), indicating that perceived support from superiors influences satisfaction with the Corona bonus (the higher the perception of support, the higher the satisfaction with the Corona bonus). Coefficient *b* (MSADCP2_Corona bonus_) = 0.33, *SE* = 0.09, *p* = 0.0008) was significant and positive, indicating that satisfaction with the Corona bonus has a positive influence on the perceived workload (the higher the satisfaction with the Corona bonus, the lower the perceived workload). On the other hand, coefficient *c’* (MSADCP7_support_) = 0.07, *SE* = 0.07, *p* = 0.3645) was not significant, so once the effect of the Corona bonus was controlled for, perceived support from superiors did not influence workload. Coefficient *c* was significant and positive (MSADCP7_support_) = 0.15, *SE* = 0.07, *p* = 0.0499), so the perception of greater support from superiors decreased the perceived workload. Finally, the *ab* effect was significant (MSADCP2_Corona bonus =_ 0.08, *SE* = 0.03, 95% *CI* [0.02–0.15], so the Corona bonus was a significant mediator between perceived support from superiors and workload; see Figure 4. 

## 4. Discussion

The results of the present study analyse the validity and consistency of the selected measurement tool and the effects of the measured stressors on the psychological stress of physicians and nurses in several hospital care units, which had more or less exposure to patients with COVID-19. In addition, the mediating role of stress-protective factors during the first health and social crisis produced by the SARS-CoV-2 pandemic was considered.

After performing the EFA, it can be stated that the psychometric properties of the instrument are valid and reliable.

Healthcare professionals are at a high risk of psychological stress. This risk increased during health crises caused by recent viral outbreaks, such as MERS and SARS-CoV-2 [29,30,31,32].

The data obtained for the Stress Scale indicate that healthcare professionals had moderate levels of stress during the first wave of the COVID-19 pandemic. For the Satisfaction Scale, satisfaction levels were high, and the analysed factors reduced the perceived levels of mental workload.

Our results show that nurses have higher levels of stress than physicians in all items of the Stress Scale, and are in agreement with most previous studies of psychological stress during a pandemic [33].

Females made up the majority of respondents in this study. This raises the possibility that a gender bias may be present. However, we have no evidence that this is the case. In addition, since the majority of health science workers in Germany are female at this point in time and this hospital is not at all untypical, any bias may simply reflect the average situation of the German population and others with a similar prevalence of females among health science professions.

From a previous meta-review of 90 articles that analyse the variable sex as a possible risk factor for mental health during different infectious outbreaks (MERS, H1N1, SARS, H7N9, etc.), 57 revealed that being a female increases vulnerability and risk of psychological distress [33]. 

Another risk factor to be taken into account, which has been analysed both in this and previous studies, is direct exposure to patients with an infectious pathology (MERS, SARS, Ebola, or N1H1). We found that emergency and ICU staff have higher levels of stress than staff working in outpatient departments, which have less exposure to COVID-19 patients. These results are in line with other studies showing that working in high-risk units due to direct contact with people carrying SARS-CoV-2 is associated with higher levels of psychological stress [29,34,35,36].

Considering the health crisis that occurs at all levels during a global pandemic, it is necessary to assess stress, both at inpatient and outpatient facilities, as both of these types of facilities are fundamental pillars of health systems and should be considered in similar studies [8].

Regarding the ever-changing management if the pandemic, as well as the initial lack of protective equipment and little or no experience of the healthcare workforce to provide care during a pandemic at this level, the results show a very weak relationship between the fear of becoming infected with SARS-CoV-2 and the lack of PPE, or healthcare management. 

In contrast, other studies have shown that a lack of information, inexperience, a shortage of PPE, or poor PPE safety are all associated with higher levels of stress, anxiety, depression, and the perception of poorer logistical support [37,38,39,40,41,42,43].

Bohlken et al. offered an explanation of their findings that can be extrapolated to this study. They pointed out that the COVID-19 pandemic did not initially affect Germany as severely as some other countries, and that the confidence in the structural and economic capacity to cope with the problems of supplying protective equipment may have been greater, which may explain the lower stress of the healthcare workforce compared to other countries [8].

Some studies show that the provision of protective equipment, the training of the healthcare workforce, clear guidelines and other appropriate resources that reduce the risk of infection also reduce psychological stress [44,45].

Although, so far, there are few longitudinal studies evaluating modifiable stress factors in healthcare professionals during the current COVID-19 pandemic, these highlight key areas that may reduce psychological stress, such as the role of social and psychological factors [33].

The results reveal that there are significant differences between nurses and physicians concerning the stress-modulating effects of protective factors on stressors. Physicians show higher satisfaction with stress-protective factors such as incentive payments or support by their superiors, which, in turn, were found to have statistically reduced stress levels. Conversely, nurses show lower satisfaction with the above measures and thus higher levels of stress. These data imply that if both groups initially had the same level of stress, the management measures and factors analysed were less effective at lowering stress in the nurses, so there is a need to focus more on improving compensation mechanisms such as psychological support or more exhaustive monitoring to detect risks early on.

Other studies have evaluated this same situation in other countries, corroborating the fact that nursing staff experience higher levels of anxiety and continuous internal stress compared to physicians [46]. Some studies point out that nurses may face more stress not only as a result of their workload and greater exposure to patients, but also because, as a predominantly female group, they face greater family burdens. This situation allows them to delegate less housework, adding and linking work stress to their daily life stress [47].

In addition, significant differences were found by department, with staff in the emergency department or ICU showing lower satisfaction with extra financial incentives or the management of the pandemic by their superiors than in other departments such as the COVID ward or the outpatient department or tent, where data revealed higher satisfaction and lower stress levels. These same results were obtained in other studies that confirm that, despite being accustomed to working with great uncertainty, the professionals who worked within these special services during the first wave of COVID felt overwhelmed by the lack of information and the great workload, which made it difficult for them to make decisions [48]. The research also pointed out that, as action protocols were established, this matter improved, but they continued to experience more stressful situations because, when in doubt, they usually provide the first and last contact with patients during their care.

Other protective factors such as increased team cohesion have been associated with reduced stress due to working in an unusual service with unfamiliar and inexperienced colleagues. Some studies support these findings, showing that positive team attitudes, intergroup relations, resilience, and psychological support decrease the perception of stress during a pandemic [11].

In our study, it was found that professionals experienced a great deal of stress and fear due to the lack of adequate protective equipment. Other authors indicate that the provision of protective material is adequate, but state that these materials are uncomfortable and usually come in one-size-fits-all format. This makes work difficult due to heat, discomfort, and difficulty of movement during working hours, thereby representing additional factors that increase feelings of exhaustion and dissatisfaction with the profession [49].

One of the main stressors that healthcare professionals exhibited, which is reflected in our study, is the fear of contagion and transmission. These results have been ratified in numerous recent studies that indicate that when workers think about the first waves that they faced without vaccines and with poor protective materials, the expression that best describes their state at the time is the fear of contagion [50]. However, the perception of the fear of contagion varies according to the previous experiences of the professionals, indicating, for example, greater stress in cases in which they have relatives who have died from COVID or in cases in which professionals passed the disease on asymptomatically [51].

The findings of this study show that along with salary increases, teamwork and communication, including with superiors, as well as the recognition of institutions and the population, are essential factors that guarantee the motivation of healthcare professionals. However, as pointed out in other studies, healthcare professionals currently feel that the recognition of their work has been neglected, which causes them to experience burnout and dissatisfaction with their profession [47].

Previous results have shown that social exclusion and avoidance as potential carriers of the virus resulted in moderate levels of stress for healthcare workers during the pandemic. Other studies have confirmed this effect [52,53,54]; it has even been proposed that social contact with the stigmatised can reduce both exclusion and stress, but this technique is not considered advisable during an epidemiological outbreak, where exclusion may actually have a protective effect [55].

To cope with the “first wave” of the pandemic, a very comprehensive learning process was necessary for all involved. Thanks to the experience gained, together with increased scientific knowledge of the virus, management plans have been progressively improved and applied to improve performance in subsequent phases of the pandemic. In this process, hospital departments have focused on improving management measures at the meso and micro levels, analysing the influence of macro-level measures.

As the most effective management measures, respondents highlighted that thorough and regular communication and information as well as cooperation between all parties involved worked particularly well in difficult and demanding situations. Strengthening psychological care services for staff is also an important measure and a service that the healthcare workers in this study assessed as necessary. In addition, they demanded more mental support from superiors and the occupational medicine department.

As previously mentioned, the deferral or cancellation of vacations is a stress factor, which could be solved by hiring more healthcare personnel and optimizing work planning, with increased rotation of shifts in an effort to maintain normal working hours to the highest degree possible.

As one of the most stressful factors was the fear of transmitting the disease to others, including family and friends, over and above the fear of infection itself, ensuring that personal protective equipment is available at all times is paramount to ensure safety and reduce stress. In addition, healthcare workers said that more frequent testing of staff was necessary, as there was little testing during the first wave. This would have reduced fear and uncertainty, as well as sick leave, due to uncontrolled infection.

Another factor that led to high levels of stress was the restriction of visits for patients who were very ill or dying, as this was ethically difficult to reconcile. Healthcare workers demanded that family members be able to communicate with patients by videoconference. In the early months of the pandemic, it was forbidden to bring any phones from healthcare staff into rooms for obvious safety and hygiene reasons, but in later stages and by ensuring thorough disinfection of the device, hospital management allowed patients to communicate with their relatives, which was a relief for both patients and healthcare staff.

The article by Sepúlveda et al. [49], like those of many other authors [8,9,10,11,12,13,15,21,22,56], limits itself to analysing stress factors, but the recommendations it provides are not based on the results obtained. Our questionnaire not only assessed specific stress factors of healthcare workers during the COVID-19 pandemic, but also incorporated items that include measures and factors that are protective of stress, such as the support perceived from both society and superiors, the existence of professional psychological support, management, and economic incentives, if any. Management has some ability to provide financial compensation, though the high-profile national bonus complicates the interpretation of the actual effectiveness of these extra incentives.

The strength of these scales is that they reliably and effectively assess the anxiety and satisfaction with management experienced by healthcare professionals, facilitating the evaluation of the Departments of Occupational Medicine dedicated to assessing their mental health. The study described herein contributes to the improvement of hospital management during health crises such as the recent COVID-19 pandemic.

Data collection took place in July and August 2020, after the end of what was considered the first wave of the COVID-19 pandemic, so the results obtained reflect the intense anxiety suffered in the exceptional situation that the professionals faced and which cannot be replicated or compared to any other of this magnitude in recent years.

Noting that this tool has been proven to be useful and given that there is no history of pandemics at this level in recent years, it could be applied not only in situations similar to the COVID-19 pandemic, but also in other local outbreaks or transient pandemics where urgent measures are required. For all its strengths, this study faces some limitations. The sample was selected non-randomly by convenience, with the number and characteristics of the subjects that comprise the sample not necessarily being representative of the total number of healthcare professionals. On the other hand, as this was a cross-sectional study instead of a longitudinal one, these is no way to determine whether the measures reduced or increased stress in the medium or long term, or even if the professionals developed other pathologies, such as depression or post-traumatic stress. Furthermore, the sample size was insufficient to carry out a confirmatory factor analysis (CFA) for both scales because, although the constructs obtained through the EFA have three or more indicators per factor, which means high precision in the estimates, the minimum required sample size is 200 subjects [25]. In addition, the study did not consider the possible impact of gender, age or years of experience, or include other healthcare providers, physiotherapists or pharmacists, who were less exposed but nevertheless subject to stress.

Future research should take into account the limitations described above. Increasing the sample size, as well as the number of health centres and their variability, will be necessary to conduct a CFA for both scales, as well as the calculation of other statistical parameters. Once the proposed instrument has been validated, longitudinal studies may provide answers to changes in stressors and how stress strategies and protective factors influence the mental well-being of healthcare workers in the medium and long term, assessing not only anxiety or psychological stress, but also post-traumatic stress or depression [49].

Data collection could also be carried out at different times during the pandemic. It would be interesting to evaluate in depth the gender variables in the anxiety experienced in hospitals and social health centres, since healthcare professions tend to have a greater female predisposition [57]. Other occupational risk factors should also be taken into account during a pandemic or similar situation in order to develop protocols and actions testing the use of new technologies like virtual reality as measures to improve diagnosis, care, and to preserve the physical and mental well-being of all healthcare workers.

Finally, since the measurement instrument was developed and administered in a hospital in Germany, it would be interesting to carry out other studies to validate the questionnaire in order to adapt and improve it. Its translation, validation and application in other countries could help to determine what differences exist between the healthcare workforces in different national health systems.

## 5. Conclusions

The pandemic caused by SARS-CoV-2 has generated and increased psychological stress in healthcare professionals. Although there is evidence of how different work and personal factors have influenced the mental well-being of healthcare workers, there is a need to develop assessment instruments adapted to this type of health crisis, increasing the reliability and consistency of the data.

Both nurses and physicians, among other healthcare staff and workers included in various studies, have shown high psychological burdens from working in care services for patients with COVID-19. Compensation mechanisms, such as incentive payments or support from superiors, are more effective for physicians than nurses, leaving the latter more vulnerable to higher levels of stress. Therefore, assessing the effectiveness of the measures implemented requires a more careful examination of nursing, as the results point to a greater need for improvement. The measures most likely to positively impact the workforce are assessing stressors and monitoring for early signs of stress to enable timely intervention

Moreover, the difference between medical services and specialties is evident, as stress is higher in the emergency department or ICU than in other departments. However, the most significant finding was that staff in the outpatient unit/screening tent for patients at the emergency doors showed the lowest stress levels, despite their exposure to the virus and the high demand for communication and coordination with the other COVID-19 units.

In this case, the heads and supervisors of the different departments continuously stressed the importance of good communication between units and this measure was the most effective at reducing stress relating to hospital logistics. This is a very important factor for management to take into account, as during the pandemic, the high demand for beds was one of the most devastating problems.

Other key factors that can modulate stress levels during an epidemiological outbreak include actions such as incentive payments, psychological support for staff in hospital facilities or adequate coverage of protective equipment and infrastructure.

The involvement of governments and institutions is essential to implement the above measures, providing the necessary resources within their means, and—above all—listening to the needs of managers and frontline healthcare workers during a health crisis.

## Figures and Tables

**Figure 1 ijerph-19-12963-f001:**
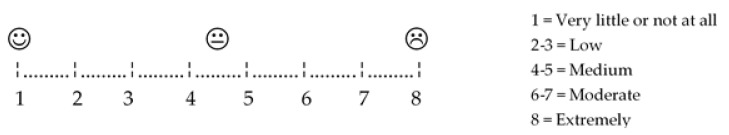
Likert-type scale with 5 score ranges. Stress Scale.

**Figure 2 ijerph-19-12963-f002:**
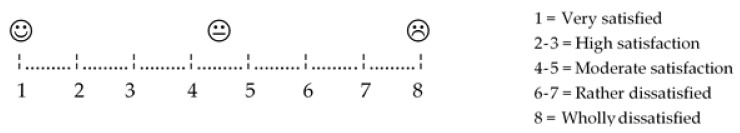
Likert-type scale with 5 score ranges. Satisfaction Scale.

**Figure 3 ijerph-19-12963-f003:**
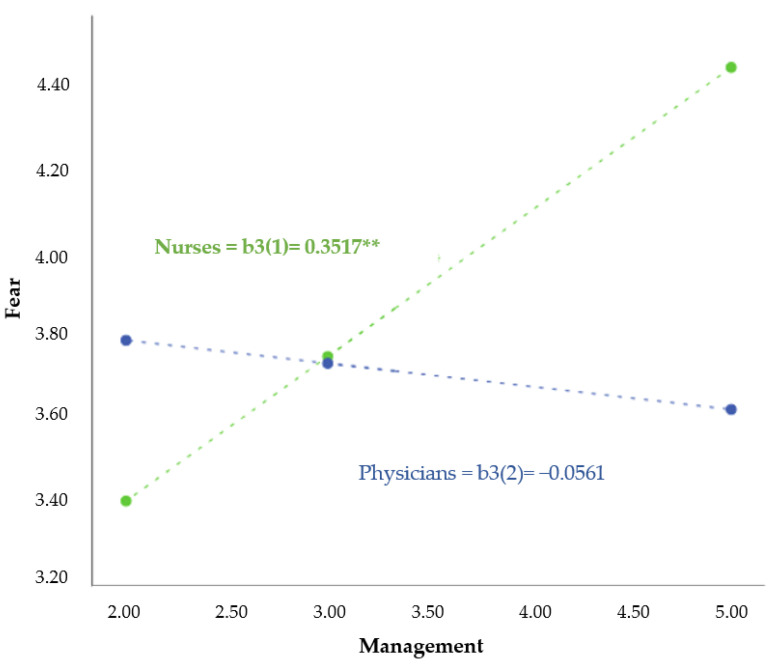
Moderation of profession variable on the effect of pandemic management over the fear of COVID-19 infection. ** *p*-value < 0.01.

**Figure 4 ijerph-19-12963-f004:**
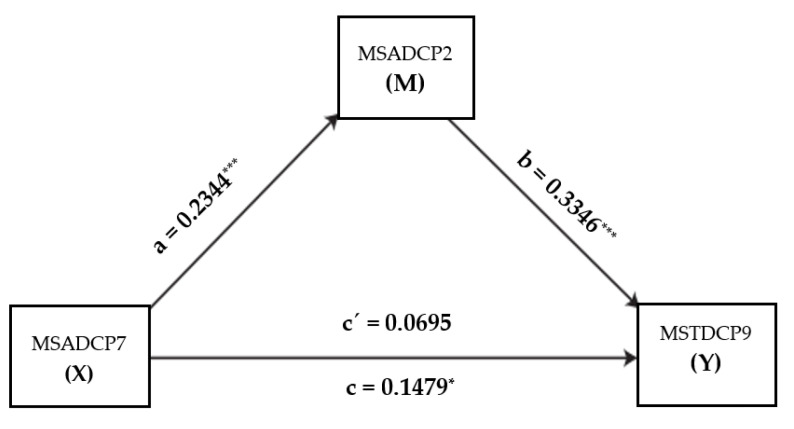
Mediation of satisfaction with the Corona bonus on the effect of the perceived support from superiors over the perceived occupational workload. * *p*-value <0.05; *** *p*-value < 0.001.

**Table 1 ijerph-19-12963-t001:** Mental stress scale during the COVID-19 pandemic (MS**T**DCP, Stress Scale).

Number	Items
1	Fear of infection
2	Fear of transmitting the infection to others
3	Use of personal protective equipment during working hours
4	Rules for visiting critically ill, terminally ill, or deceased patients due to COVID-19
5	The dynamics of the pandemic due to frequent changes in protocols
6	Lack of protective equipment
7	Working in an unfamiliar area with unfamiliar and inexperienced colleagues
8	Fear of social exclusion as potential carriers of SARS-CoV-2 due to the profession
9	The changed and increased workload
10	The extra time burden
11	The extra private burden stemming from social isolation measures, school closures, etc.

**Table 2 ijerph-19-12963-t002:** Mental satisfaction scale during the COVID-19 pandemic (MS**A**DCP, Satisfaction Scale).

Number	Items
1	Satisfaction with the management of the pandemic
2	Incentive payment (“Corona bonus”) ^1^
3	Appreciation experienced from institutions
4	Appreciation experienced from the public
5	The care provided by the occupational medicine department
6	The availability of a psychological counselling helpline
7	Support and attention from superiors in the health centre
8	Teamwork
9	Information management
10	Attention from the Hygiene Department of the centre

^1^ Incentive payments during the pandemic in Germany were called the “Corona bonus”.

**Table 3 ijerph-19-12963-t003:** Selected comments from the Stress Scale.

“The correct protection was only partial, as for a few days they recommended wearing FFP2 for longer than appropriate, increasing the risk of infection and illness”. “They didn’t ask if you wanted to work in the Covid area, they didn’t give us any other option”. “In addition to the fear of contagion, the time without contact with family members” “Children with asthma at home, and my mother” “A lot of discomfort from the personal protective equipment (PPE).” “They made me sweat a lot” “The lack of oxygen was overwhelming” “I did not feel it was right that at the beginning it was not possible to visit the deceased” “It often was our decision” “Working with unfamiliar colleagues was stressful” “At first it was very scary to work in that ward” “There was no lack of PPE at any time” “There was no lack of material, but we used it for too long” “The potential risk of infection and thus not having contact with family and friends for months was very hard” “My partner was afraid” “I had to increase my working hours from 50 to 75%” “The private extra burden was higher, due to having to take care of the children” “Lockdown + working with Covid + closed schools = more pressure and stress”

**Table 4 ijerph-19-12963-t004:** Selected comments from the Satisfaction Scale.

“In the beginning, things were not managed well at all, then better” “What management?” “I don’t think it’s right that all healthcare workers should have the same Corona bonus” “Those working directly with Covid should have received more” “Clapping doesn’t add a single euro to the bank account” “I don’t believe in the duration of this recognition by society” “We should have had regularly scheduled PCR checks” “I didn’t know there was a psychological helpline” “Nursing and administration management was not seen at any time” “Support was not sustained” “Support from the supervisor was enormous” “I didn’t feel the support of the public, but I did feel more recognition from fellow doctors”

**Table 5 ijerph-19-12963-t005:** Internal consistency of the Stress Scale.

Item	*M* ^1^	*Var* ^2^	*ITC* ^3^	*A* ^4^
**1/MSTDCP 1**	46.31	182.33	0.49	0.81
**2/MSTDCP 2**	44.45	183.84	0.44	0.82
**3/MSTDCP 3**	45.65	185.15	0.40	0.82
**4/MSTDCP 4**	45.35	183.41	0.42	0.82
**5/MSTDCP 5**	45.28	179.50	0.60	0.81
**6/MSTDCP 6**	45.04	184.53	0.37	0.83
**7/MSTDCP 7**	47.15	180.05	0.47	0.82
**8/MSTDCP 8**	46.18	176.38	0.51	0.81
**9/MSTDCP 9**	45.88	173.79	0.64	0.80
**10/MSTDCP 10**	45.68	172.53	0.64	0.80
**11/MSTDCP 11**	45.31	176.50	0.56	0.81

^1^*M* = mean (deleted item). ^2^ *Var*. = variance of the scale (deleted item). ^3^ *ITC* = item-total correlation. *A*
^4^ = Cronbach’s alpha (deleted item).

**Table 6 ijerph-19-12963-t006:** Internal consistency of the Satisfaction Scale.

Item	*M* ^1^	*Var* ^2^	*ITC* ^3^	*A* ^4^
**1/MSADCP1**	46.31	182.33	0.49	0.81
**2/MSADCP 2**	44.45	183.84	0.44	0.82
**3/MSADCP 3**	45.65	185.15	0.40	0.82
**4/MSADCP 4**	45.35	183.41	0.42	0.82
**5/MSADCP 5**	45.28	179.50	0.60	0.81
**6/MSADCP 6**	45.04	184.53	0.37	0.83
**7/MSADCP 7**	47.15	180.05	0.47	0.82
**8/MSADCP 8**	46.18	176.38	0.51	0.81
**9/MSADCP 9**	45.88	173.79	0.64	0.80
**10/MSADCP 10**	45.68	172.53	0.64	0.80

^1^*M* = mean (deleted item). ^2^ *Var.* = variance of the scale (deleted item). ^3^ *IQ-T* = item-total correlation. *A*
^4^ = Cronbach’s alpha (deleted item).

**Table 7 ijerph-19-12963-t007:** Stress Scale rotated component matrix.

	Component ^a^		
Item	1	2	3
MSTDCP9	0.804		
MSTDCP10	0.0771		
MSTDCP3	0.716		
MSTDCP11	0.586		0.480
MSTDCP6		0.797	
MSTDCP5	0.353	0.659	
MSTDCP4		0.627	
MSTDCP7		0.613	
MSTDCP8	0.368	0.464	
MSTDCP2			0.841
MSTDCP1			0.826

Extraction method: principal component analysis. Rotation method: Varimax with Kaiser normalization ^a^.

**Table 8 ijerph-19-12963-t008:** Structure of the Stress Scale after the EFA.

Factor	MSTDCP Items	α	Definitive Dimension
F1	3, 9, 10	0.75	Mental stress arising from workload
F2	4, 5, 6, 7, 8	0.72	Mental stress due to constant changes in work, protocols, and availability of material/infrastructure
F3	1, 2, 11	0.71	Fear of infection, transmission, and additional private burden due to social isolation measures

**Table 9 ijerph-19-12963-t009:** Stress Scale and dimensions.

Number	Dimension	Item
1	D3	Fear of infection with COVID-19
2	D3	Fear of transmitting the infection to others
3	D1	Wearing PPE
4	D2	Visiting guidelines for critically ill or dying COVID-19 patients
5	D2	Pandemic dynamics due to changing standards/guidelines
6	D2	Temporary lack of PPE
7	D2	Working in an unfamiliar speciality with non-expert colleagues
8	D2	Being excluded by others as a potential carrier of infection due to working in a COVID-19 area
9	D1	The workload in terms of content
10	D1	The workload in terms of time
11	D3	The extra private burden

**Table 10 ijerph-19-12963-t010:** Satisfaction Scale rotated component matrix.

	Component ^a^	
Item	1	2
MSADCP10	0.818	
MSADCP9	0.799	
MSADCP1	0.774	
MSADCP7	0.712	
MSADCP6	0.689	
MSADCP8	0.648	0.321
MSADCP5	0.570	
MSADCP4		0.859
MSADCP3	0.332	0.800
MSADCP2	0.372	0.572

Extraction method: principal component analysis. Rotation method: Varimax with Kaiser normalisation ^a^.

**Table 11 ijerph-19-12963-t011:** Structure of the Satisfaction Scale after the EFA.

Factor	Items	α	Definitive Dimension
F1	1,5,6,7,9,10	0.85	Mental satisfaction associated with management and the measures taken for physical security and psychological protection
F2	2,3,4,8	0.70	Mental satisfaction related to perceived support and extra financial compensation

**Table 12 ijerph-19-12963-t012:** Satisfaction Scale and dimensions.

Number	Dimension	Item
1	D1	Coronavirus crisis management
2	D2	Corona bonus/incentive payment
3	D2	The experienced appreciation
4	D2	Public support
5	D1	Care provided by the occupational medicine department
6	D1	Availability of a psychological support hotline
7	D1	With the psychological/mental support of my superiors
8	D2	Team cohesion
9	D1	With information management
10	D1	With the supervision of the specialised hygiene centre

**Table 13 ijerph-19-12963-t013:** Relationships between the scales and subscales of the exploratory factor analysis.

	Stress Scale	F1-MSTDCP	F2-MSTDCP	F3-MSTDCP	Satisfaction Scale	F1-MSADCP	F2-MSADCP
**Stress Scale**	1						
**F1-MSTDCP**	0.76 **	1					
**F2-MSTDCP**	0.85 **	0.52 **	1				
**F3-MSTDCP**	0.76 **	0.48 **	0.46 **	1			
**Satisfaction Scale**	0.37 **	0.31 **	0.35 **	0.24 **	1		
**F1-MSADCP**	0.34 **	0.26 **	0.34 **	0.23 *	0.95 **	1	
**F2-MSADCP**	0.35 **	0.35 **	0.24 **	0.25 **	0.85 **	0.63**	1

* *p* ≤ 0.05, ** *p* ≤ 0.01.

**Table 14 ijerph-19-12963-t014:** Differences in the means according to the variables of the department for the Stress Scale.

Department	*n*	Mean	SD
**Emergency Department (ED)**	34	52.47	14.88
**Intensive Care Unit (ICU)**	48	50.77	12.39
**COVID Ward/Station**	77	50.92	15.31
**Outpatient Tent**	17	41.11	14.74

**Table 15 ijerph-19-12963-t015:** Differences in the means according to the variables of the department for the Satisfaction Scale.

Department	*n*	Mean	SD
**Emergency Department (ED)**	23	35.82	17.30
**Intensive Care Unit (ICU)**	38	28.53	10.52
**COVID Ward/Station**	51	27.70	11.69
**Outpatient Tent**	1	26.50	11.78

## Data Availability

The data presented in this study are openly available from [Zenodo] at [doi.org/10.5281/zenodo.5963418] [doi.org/10.5281/zenodo.6329978] (accessed on 8 March 2022). Readers are advised to contact the main researcher, who will be able to explain the variables, since she speaks English, German, and Spanish.

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
