# Peer review of "Causes of Stress among Healthcare Professionals and Successful Hospital Management Approaches to Mitigate It during the COVID-19 Pandemic: A Cross-Sectional Study"

_ijerph, 2022, doi:10.3390/ijerph191912963_

Round 1

Reviewer 1 Report (Previous Reviewer 3)

The authors allocated substantial effort to improving the quality of the manuscript and I have no problem in recommending it for publication. 

Author Response

Reviewer 2 Report (New Reviewer)

The paper

Causes of stress among healthcare professionals and success-ful hospital management approaches to mitigate it during the Covid-19 pandemic

has incorporated a large number of corrections, which has increased the quality of the paper for publication in IJERPH.

However, I believe that authors should be careful with the aesthetics and presentation of the paper.

I recommend improving the following tables and figures:

Tables: 1, 2, 3,4,7,8,9,10

Figure: 1, 2, 4

Tables with the contents aligned to the left

The figures, cleaner, with better design

Author Response

Reviewer 3 Report (New Reviewer)

This is an interesting study which aims to explore the most important sources of stress and which of the protective measures applied to counteract it among healthcare staff . Title

1.     Add study design

2.     Capitalize first letter in each word.

Abstract

Add the words (background, objectives, methods, results and conclusion) so the abstract be structured. Add date of data collection.

Introduction:

1.     The introduction is good but very long. Try as possible to remove paragraphs that are not directly-related to the topic. i.e. paragraphs number 3 and 4.

2.     The following paragraph could move from the introduction to the methods (Procedures and instruments) section:

“The department of occupational medicine developed an ad hoc questionnaire to detect which problems influence the mental health of healthcare staff working in Covid-19 patient care services. The questionnaire is divided into two clearly defined scales, one with 11 items and the other with 10 items. As such, ………as well as the actions that can be most effective in reducing it.”

3.     Give a brief definition of ad hoc instruments with first time you have mentioned.

4.     References number 14, 21 and 20 are more than 10 years old, try to update them.

Methods:

5.     In participant section only total number of healthcare provider (HCP) should be given. The other details like gender and type of the HCP should move to the result section.

6.     How did you get with this number (291)? Are they all the HCP available in the hospital or did you do a sample size calculation? It is not clear in your writing.

7.     The following paragraph should move to the (statistical analysis) section : “The data were grouped, coded and analyses using the IBM SPSS 25 statistical analysis software and the PROCESS v3.1 macro for IBM SPSS v24 [26], to conduct mediation and moderation analysis.”

8.     You have mentioned that the questionnaire was divided in to two sections three times in the method section. Please mention that one time.

9.     Way only nurses and physicians as HCP were included? What about pharmacists and other HCPs?

Results

10.  Check abbreviations like PPE is written in full with first use in the text. It was written in fill in the table only.

Discussion:

11.  References number 27, 30, 39, 50 and 60 are more than 10 years old, try to update them.

12.  Non of the HCP-related factors were considered i.e. gender, age, years of experience... This can be added to the limitation.

Author Response

This manuscript is a resubmission of an earlier submission. The following is a list of the peer review reports and author responses from that submission.

Round 1

Reviewer 2 Report

I believe this paper is very interesting. There is not only theoretical but practical value behind it. In addition, I need to underline that I specialize in qualitative methods and because of this I think that the paper must be reviewed by a statistician

Reviewer 3 Report

>> The language usage throughout this paper need to be improved, the author should do some proofreading on it. Give the article a mild language revision to get rid of few complex sentences that hinder readability and eradicate typo errors.
>> Your abstract does not highlight the specifics of your research or findings. Rewrite the Abstract section to be more meaningful. I suggest to be Problem, Aim, Methods, Results, and Conclusion.
>> Introduction section can be extended to add the issues in the context of the existing work and how proposed algorithms/approach can be used to overcome this.
>> Add main contributions list as points in the Introduction section.
>> Add the rest organization section at the end of the Introduction section.
>> More clarifications and highlighted about the research gabs in the related works section.
>> identified research gaps and contribution of the proposed study should be elaborated.
>> The authors should highlight shortcomings of the existing solutions in the related work section;
>> The authors should consider more recent research done in the field of their study Such as: 
        1) Automated System for Identifying COVID-19 Infections in Computed Tomography Images Using Deep Learning Models. Journal of Healthcare Engineering
          2)  A Multi-Agent Deep Reinforcement Learning Approach for Enhancement of COVID-19 CT Image Segmentation. Journal of personalized medicine                 
          3) COVID-19 anomaly detection and classification method based on supervised machine learning of chest X-ray images. Results in Physics
         4) Review on COVID‐19 diagnosis models based on machine learning and deep learning approaches. Expert Systems
         5) Artificial intelligence-based solution for sorting COVID related medical waste streams and supporting data-driven decisions for smart circular economy practice. Process Safety and Environmental Protection
>> I feel that more explanation would be need on how the proposed method is performed.
>> If no one has proposed before a method like the proposed algorithm, this claim should be highlighted much more. Else, it should be indicated who has done this, and it should be indicated what the innovations of the current paper are.
>> Authors should add the parameters of the algorithms.
>> A comparison with state of art in the form of table should be added
>> Results need more explanations. Additional analysis is required at each experiment to show the its main purpose.
>> The results are not easy to follow. The proposed method and experiments are not clearly illustrated. Result and Discussion section is inadequate. Need more attention and better explanation.
>> The authors should further detail the preparation of the dataset.
>> The Limitations of the proposed study need to be discussed before conclusion.
>> Please include the contribution of this paper, limitation of this study and future research in the Conclusion Section.

Round 2

Reviewer 1 Report

I would like to thank the authors for their detailed reply. 

However, despite the changes made to the paper, I do maintain that the founding problem remains: I do not believe it to be scientifically sound to propose a new clinical scale validation with a 198 people sample, across a 2-month period evaluation. It is a methodological problem which diminishes the greatest interest of the paper, which would have been a validation of a scale to evaluate stressors and protective factors against COVID in healthcare workers. It would also have been interesting to gather prospective data on this matter. 

I do believe there is potential in this paper which is not being exploited to its maximum. I suggest a methodological revision which would bring greater scientific value to the paper and interest to the healthcare community. 
